# Assessing the quality of training programs for teachers of students with disabilities in light of recent trends in the Kingdom of Saudi Arabia

**Rasheed Khuwayshan Algethami** [ORCID] *

Department of Special Education, College of Education, Taif University, Taif, Kingdom of Saudi Arabia

* rkgethami@tu.edu.sa

**Data Availability Statement:** The data relevant to this study are available from figshare at https://doi.

## Abstract

The study aimed to assess the quality of training programs for teachers of students with disabilities in light of recent trends in the Kingdom of Saudi Arabia. The descriptive-inferential approach was used in this study, which was applied to male and female teachers who attended an initiative of training teachers of students with disabilities. The initiative was launched by the Ministry of Education in the summers of 2019/2020. The sample of the study consisted of (272) teachers who were chosen by the stratified random sampling method from three cities (Jeddah, Makkah, and Taif). To achieve the objective of the study, a questionnaire was used to collect data after checking the issues of validity and reliability indications. The results of the study revealed that the overall degree of the assessment of the quality of training programs for teachers of students with disabilities in light of recent trends from the point of view of the study sample was medium. Also, there were statistically significant differences in the assessment degree of the quality of training programs for teachers of students with disabilities in light of recent trends due to academic qualification in favor of postgraduate studies, and years of experience for those with more than 10 years of experience. However, no statistically significant differences were shown due to gender. In light of these results, the study presented some recommendations. There were weaknesses in the training programs in terms of training teachers on their various roles and responsibilities, standards for the effective use of educational skills, and the development of values and ideas. Also, these training programs did not take into account the development of the values associated with technology and its relationship to teachers. In light of the results, the study recommended that the Ministry of Education should pay attention to the adoption of a strategic plan to improve the quality of training programs for the teachers of students with disabilities by relying on modern models and trends in teacher training. I addition, there is a need to conduct a diagnostic study of the most important challenges and obstacles and proposed solutions to the quality of training programs for the teachers of students with disabilities.

## Introduction

Training and professional development for teachers of students with disabilities are among the modern approaches that occupy educational institutions and those interested in reform issues

org/10.6084/m9.figshare.20072210.v1 (10.6084/m9.figshare.20072210).

**Funding:** This study was funded by the Deanship of Scientific Research at Taif University, Kingdom of Saudi Arabia (Grant No. 1-1442-17) awarded to Rashid Khuwayshan Algethami. The funders had no role in the study design, data collection and analysis, the decision to publish, or the preparation of the manuscript.

**Competing interests:** he authors have declared that no competing interests exist.

in special education at present. They contribute to enhancing the knowledge, skills, and experiences of teachers in various fields emotionally and professionally so that they do not stop at the level they reached upon graduation or employment in the educational sector [1].

In recent years, the term (Quality) has spread. It is a modern management philosophy based on a number of modern directed management concepts on which basic administrative means and creative efforts are mixed with specialized technical skills to raise the level of performance and continuously improve and develop institutions and human resources. Therefore, there have been voices calling for quality in the education sector similar to other sectors. It was a turning point in the path of educational reform through the presence of a team with extensive experience to lead this movement and change and achieve the objectives of improving educational quality in pre-service and in-service teacher training programs [2].

Quality is of great importance in determining the basic pillars in the framework of practical application in public education institutions. These pillars would indicate the basic facts that to be relied upon in the field of quality assurance and quality control to apply advanced methods to ensure their standards as well as continuous improvement and achieve the highest possible levels in educational practices and processes. Thus, the outputs of educational institutions are improved. One of the most important of these pillars is training and professional development for teachers [3].

Therefore, the researcher believes that the issue of quality in training programs for teachers constitutes a challenge for public and private education institutions. The initiative to raise this topic is important in drawing the attention of those in charge of training programs seriously. It is necessary to strive continuously to develop teachers' skills, including teachers of students with disabilities scientifically and professionally. The continuous development of teachers' skills will keep pace with the era of innovation, spread the culture of training as a basis for entrepreneurship, and develop training strategies in line with modern global trends in training.

## Statement of the problem

Despite the great efforts that the training process for teachers of students with disabilities has achieved at the local level in the Kingdom of Saudi Arabia, some challenges and difficulties still stand in the way of achieving the desired goals of training. This has been confirmed by the results of the survey conducted by the current researcher. Individual interviews were conducted with 20 teachers of students with disabilities. They indicated a disparity and controversy about the effectiveness of the training programs provided to the teachers of students with disabilities in terms of attendance and participation in such programs. Also, there were differences in their motivation and personal beliefs towards training programs as a result of weaknesses in novelty or the lack of keeping pace with modern educational developments, or connection with their community and new technical competencies. Accordingly, the need for this study has emerged and aimed to answer the following questions:

1. What is the assessment level of the quality of training programs for the teachers of students with disabilities in light of recent trends?

2. Are there statistically significant differences in the assessment level of the quality of training programs for the teachers of students with disabilities in light of recent trends due to the gender variable?

3. Are there statistically significant differences in the assessment level of the quality of training programs for the teachers of students with disabilities in light of recent trends due to the variable of academic qualification?

4. Are there statistically significant differences in the assessment level of the quality of training programs for the teachers of students with disabilities in light of recent trends due to the variable of years of experience?

## Significance of the study

The significance of this study is evident by focusing on an important topic of training and human resources. It focuses on the assessment of the quality of training programs for the teachers of students with disabilities in light of recent trends as well as the enrichment of the global and human library with the literature of the theoretical study and its results. It is hoped that the results of this study will benefit the Ministry of Education, the General Department of Special Education, and other departments of special education in various regions of the Kingdom of Saudi Arabia through the information provided by the study on assessing the quality of training programs for teachers of students with disabilities in light of recent trends. This information may improve the work of training and professional development centers for the teachers of students with disabilities. Also, it is hoped that those responsible for developing educational policies and training programs for the teachers of students with disabilities will benefit from the results of this study in obtaining a comprehensive vision of the reality of those programs and making them aware of their weaknesses and strengths to develop appropriate solutions for their development and to spread their benefit to the teachers of students with disabilities. In addition, they will contribute to developing the work of teacher training centers in general and teachers of students with disabilities, in particular, to be aware of the requirements of training programs and their assessment to be reflected in their future performance.

## Objectives of the study

This study aimed to determine the level of assessment of the quality of training programs for the teachers of students with disabilities in light of recent trends to work for their development and enhancement to achieve effectively their desired goals. Also, the study aimed to unveil any statistically significant differences in the study sample's responses about whether the level of assessment of the quality of training programs for the teachers of students with disabilities in light of recent trends is due to gender, academic qualification, and years of experience to see their views and develop the quality level of training programs for teachers of students with disabilities in line with recent trends to achieve competitive advantage.

# Background of the study

Human resources have received great attention from departments and institutions because they impact the success of these bodies in various administrative, social, and economic fields. Therefore, the issue of developing the competencies of human resources in these bodies and institutions is one of the matters becoming imperative for their advancement and the services they provide to society. Also, the role of training is considered one of the most important processes in the development of human resources in these bodies and institutions [4]. The training philosophy is based on a clear logic that assumes that the required numbers and types of human competencies to practice certain jobs require a high level of competence as is the case of the required disciplines [5]. The philosophy of training is based on a basic rule that even in the case of academic institutions from which specialists in various fields of work graduate, they will still need training. Training programs provide methods and open new horizons of professional and scientific knowledge for the practitioner, whose theoretical study allows him to interact with them. In addition, the practitioner is in a constant need to renew his information,

develop existing knowledge, and stand up to the latest methods and theories in the field of specialization. Organized training programs provide the above-mentioned skills to PR actioners [6]. On the approach of the various community institutions, the educational institutions proceeded and devoted attention to training their employees, especially teachers, including teachers of students with disabilities [7]. This came as a result of the importance of professional development and professional growth to provide the teacher with various knowledge, skills, and experiences to keep pace with modern educational developments in their field of specialization in light of their needs [8]. In-service teacher training is an important factor for developing their performance and increasing productivity. Therefore, it is considered an investment expenditure that achieves a tangible return to contribute to meeting the needs of the economic and social growth of the nation. Training represents the satisfaction of teachers' psychological and cognitive needs reflected in the mastery of their work for the sake of professional growth [9].

Teacher preparation and training programs are important in their professional development and growth. They are planned and organized programs according to educational and psychological theories. Also, they are carried out by specialized educational institutions to provide teachers with scientific and applied experiences that enable them to grow in the teaching profession and increase their educational production [3].

Interest in developing teacher training programs and ensuring their quality is one of the most important contemporary global trends. They receive increasing attention with the different degrees of focus on the development processes depending on the societal context from one country to another. Interest in teacher preparation and development has become a political priority in most countries of the world [10]. In developed countries, there is an increasing interest to ensure the provision of sufficient and required numbers of teachers to meet the increasing retirement cases while ensuring the quality of preparation. In developing countries, we find an increasing need to raise the level of qualifications of teachers with attention to quality. In light of contemporary educational trends and the emergence of new patterns and methods in teaching, it becomes clear that the urgent need for changing the teacher's future roles. Consequently, teacher preparation and training programs are assessed in light of contemporary roles and challenges. For these programs to be effective, they require development in their objectives, mechanisms, and methods to overcome their present weaknesses [11].

Hence, educational institutions seek to reach the quality of service and the scientific and professional levels of the teacher by improving his educational, research, and career performance by setting up special training programs. This can be implemented and supervised by experienced and specialized competencies in the field of training. As a result, the interest in training programs for teachers has emerged. Training programs and courses equip teachers with new ways to facilitate the teaching process and help them to overcome any obstacles by the development of the educational field. Also, they facilitate the effort made to communicate ideas to students and ways to memorize and review them [12]. Therefore, training and professional development programs for teachers must include various areas that include mastering modern teaching strategies and enhancing teachers' competencies in preparing lessons in a manner commensurate with learners' needs. Also, they include using modern technology in the teaching process in the classroom [3]. The importance of teacher training programs including the teachers of students with disabilities and recent trends has emerged and aimed at achieving teachers' competencies and raising their level of performance in this changing world. They also emphasize the continuity of this performance effectively when working professionally. In addition, some of these trends emphasize practical aspects and give teachers special weight in preparation and training programs [13]. Hence the need for educational systems to consider developing programs to train teachers of students with disabilities continuously and thus keep pace with the rapid civilizational developments taking place in the era [8].

Therefore, several trends have emerged in teacher training programs focusing on different models, approaches, processes, and competencies [14]. Some of these approaches are based on competency. It focuses on the importance of training teachers to possess skills such as planning education, selecting and organizing content, using methods and means that achieve goals and assessment, the teacher's role, and various responsibilities as a guide and leader. Another approach includes teacher training programs that are based on science, technology, and society. Bybee [15] provided a conceptual framework for the knowledge, skills, and values of scientific and technical enlightenment. He defines a set of standards that can be used in building training programs such as standards for acquiring scientific and technical knowledge by studying life matters, and contemporary social issues. They also include standards for the effective use of educational skills that depend on scientific and technical investigation, participation in information gathering, problem-solving, developing values and ideas related to science and technology and their relationship to society. The current study focused on these two approaches because they are among the most recent approaches that keep pace with the present era and its educational, scientific, and technical developments in the field of special education and teacher preparation and training.

The teacher of students with disabilities is an important hub in the teaching and learning processes of individuals with disabilities. He is responsible for implementing the individual educational plan, developing their knowledge and skills, and following up on their achievements and performance [16]. Likewise, he is the main executor of the individual educational program, follow-up to the family plan, the guide for students with disabilities in the processes of teaching and learning, and responsible for following up on their academic achievement and solving their problems [17]. Training the teachers of students with disabilities is one of the issues that concern educational officials who believe in their great role in educating students with disabilities. Also, they refine their skills to perform well in teaching students with disabilities and keep pace with the scientific and technical development in the education of students with disabilities [18]. Hence, specifically in the summers of 2019 and 2020, the Kingdom of Saudi Arabia, represented by the Ministry of Education and the General Administration of Special Education in cooperation with a group of companies, associations, and experts in the field of training and human development and in various public educational areas launched an initiative to train special education teachers (teachers of students with disabilities). The initiative aimed to provide teachers with strategies, teaching methods appropriate for teaching students with disabilities according to categories and grades, discovering problems among children with disabilities, and ways to deal with them. Also, it aimed to train teachers on using appropriate assistive technologies in teaching and learning students with disabilities, monitoring practices based on evidence and scientific evidence in the field of disability. In addition, it targeted providing teachers with the teaching methods of students with disabilities in light of the COVID-19 pandemic and ways how to create an appropriate psychological atmosphere for children with disabilities using storytelling and social story [19].

As mentioned above, the significance of this study is evident by focusing on an important topic of management and human resources in educational institutions. The study focuses on the issue of the quality of training programs for teachers of students with disabilities in light of recent trends.

## Methods

In this study, the descriptive-inferential approach was used by following the survey method because of its appropriateness for the nature of this study. The study instrument was distributed to the study sample to answer the research question and thus the objectives.

## Population and sample of the study

The study was applied to (860) male and female teachers who attended an initiative for training the teachers of students with disabilities. The initiative was launched by the Ministry of Education in the summers of 2019/2020. The sample of the study consisted of (272) teachers from three cities (Jeddah, Makkah, and Taif) in the first semester of the year 2021–. The sample constituted (31.6%) of the study population. The sample size was determined based on [20]. It was selected by the stratified random sampling method and was distributed according to the variables of the study: gender, academic qualification, and years of experience. Table 1 shows the distribution of the study sample according to the variables of the study.

## Instrument of the study

In this study, a questionnaire was used to collect data on the assessment level of the quality of training programs for the teachers of students with disabilities in light of recent trends in the Kingdom of Saudi Arabia. Previous studies related to the quality of teacher training programs have been referred in this study to in light of recent trends [3, 8, 15]. These studies were benefited in identifying the domains for assessing the quality of training programs in light of recent trends, and then appropriate items were formulated for the domains in light of recent trends. The instrument, in its final version, consisted of (35) items distributed into seven domains after verifying its indications of validity and reliability. The first domain, planning for education, had six items. The second domain, content selection and organization, included four items. The third domain, methods and strategies for achieving goals and assessment, had seven items. The fourth domain, the role of the teacher and his various responsibilities, consisted of six items. The fifth domain, standards for acquiring scientific and technical knowledge, contained four items. The sixth domain, standards for the effective use of educational skills, had four items. Finally, the seventh domain, standards for developing values and ideas, consisted four items. To interpret the respondents' answers, they were asked to put a tick ($\sqrt{}$) in front of each of the domains' items using a five-Likert scale (very high, large, medium, low, and very low). For correcting the instrument, the following values (5, 4, 3, 2, 1) were given to the aforementioned degrees.

 **Validity of the study instrument.** The validity of the study instrument was verified by following two methods:

a. Face validity: To verify the content of the study instrument in its initial version, it was reviewed by ten experts of faculty members specialized in special education and human resources in Saudi universities. They were asked to ensure the accuracy of the linguistic formulation and the suitability of the instrument to achieve the objectives of the study. In light of the experts' opinions and suggestions, the required amendments whose importance percentage reached (80%) were considered. The most prominent observations of the experts

**Table 1. Frequencies and distribution of the study sample according to the variables of the study.**

| Variable | Category | No. |
|---|---|---|
| Gender | Male | 113 |
| | Female | 159 |
| Academic qualification | Bachelor | 182 |
| | Postgraduate studies | 90 |
| Years of experience | 1–10 years | 135 |
| | Above 10 years | 137 |
| | Total | 272 |

were the linguistic reformulation of some items in terms of modifying some words and vocabulary to make the item clearer and more measurable. Finally, the study instrument was produced in its final version and consisted of (35) items.

b. Internal consistency: To verify the internal consistency of the study instrument, it was applied to a sample of (25) male and female teachers who were chosen from the outside of the study sample. The Pearson correlation coefficient was calculated between the item and the domain to which it belongs. The values ranged from (0.31–0.55). Also, the Pearson correlation coefficient was measured between the item and the total score of the instrument. The values ranged from (0.33–0.87). Accordingly, the instrument was considered valid for what it was prepared for.

**Reliability of the study instrument.** The reliability of the study instrument was verified by two methods. The first method was the test-retest method. It was applied to a sample of teachers selected from the outside of the study sample (25), and then re-applied to the same sample with a two-week period. The Pearson correlation coefficient was calculated between the scores of respondents on the instrument in both applications and each of its domains. The second method was Cronbach's Alpha. The reliability coefficient for the instrument as well as the domains was calculated. Table 2 shows the results.

Table 2 showed that the reliability coefficients by the test-retest method of the study instrument domains ranged between (0.90–0.95) and scored (0.90) on the total instrument. Also, the reliability coefficients using the internal consistency method "Cronbach's alpha" for the domains ranged between (0.87–0.97) and scored (0.91) on the whole instrument. These values are suitable and indicate that the study instrument has a high degree of reliability.

## Procedures of the study

To achieve the objectives of the study, the study problem and its elements were identified. Then, the official sources in the education departments of Jeddah, Makkah, and Taif were referred to determine the study population. The sample of the study was calculated, and the method of selection was then identified. Then, the study instrument was prepared and its validity and reliability were verified in the Saudi Arabian context. After that, a written ethical approval was obtained from the Scientific Research Ethics Committee of the Deanship of Scientific Research at Taif University, Saudi Arabia addressed to the educational departments in Jeddah, Makkah and Taif regions to facilitate the researcher's task in applying the study instrument. Moreover, permission was obtained from school principals after being informed of the researcher's task to implement the study instruments. Then, the instrument was distributed to the target sample electronically using an electronic link via Google Form, using (WhatsApp)

**Table 2. Test-retest and Cronbach's Alpha coefficients of the study instrument.**

| No. | Domain | Test-retest | Cronbach's Alpha |
|---|---|---|---|
| 1 | Planning for education | 0.92 | 0.89 |
| 2 | Content selection and organization | 0.90 | 0.87 |
| 3 | Methods and strategies for achieving goals and assessment | 0.94 | 0.97 |
| 4 | The role of the teacher and his various responsibilities | 0.93 | 0.90 |
| 5 | Standards for acquiring scientific and technical knowledge | 0.90 | 0.93 |
| 6 | Standards for the effective use of educational skills | 0.93 | 0.91 |
| 7 | Standards for developing values and ideas | 0.95 | 0.92 |
| | Total | 0.90 | 0.91 |

due to the conditions of the Coronavirus (Covid-19) and the necessity to adhere to the precautionary measures of not trading papers to protect others and not transmit infection. The study instrument was applied to the study sample after explaining the instructions and how to respond to the instrument items. The consent of study sample was informed in a written form. A link of the consent pdf file was included at the beginning of the electronic questionnaire. The participants should click the link, read the consent, and if they agree to the terms, they press 'agree' to participate in the study. The consent included the title of the study, purpose of the study, role of participants, agreement to fill the questionnaire, confidentiality of data, use of data for only this purpose of the study. Also, the respondents were sent letters of approval to facilitate the researcher's task. Then, the data were collected, checked, and analyzed using SPSS. After that, the results were extracted, organized in tables, and reported. Finally, the results of the study were discussed, and recommendations were suggested. To judge the level of the means of the items and domains of the study instrument and the instrument as a whole, the statistical standard was used as follows:

Range of degree = maximum value- minimum value÷ number of options.

Range of degree = 5–1 = 4÷5 = 0.8. Therefore, the statistical standard is as shown in Table 3.

## Statistical processing

This study used the statistical software of (SPSS) version (23) to analyze the data of the study and answer the research questions. Pearson correlation coefficient to verify the consistency of the instrument and Cronbach's alpha coefficient to verify the reliability of the study instrument. Also, means, standard deviations, and ranks were calculated to answer the first question to identify the level of assessment of the quality of training programs for teachers of students with disabilities in light of recent trends from the point of view of the study sample. In addition, the t-test was used to answer the second, third, and fourth questions to show the significance of the differences between the means for the level of quality assessment of training programs for teachers of students with disabilities in light of recent trends from the point of view of the study sample according to demographic variables (gender, academic qualification, and years of experience).

## Results

This section presents the results that have been reached. The results are reported according to the sequence of the research questions.

Results of the first research question: *What is the assessment level of the quality of training programs for the teachers of students with disabilities in light of recent trends*?

For answering this question, the means and standard deviations of the participants' responses on the assessment level of the quality of training programs for students with disabilities in light of modern trends. Table 4 displays the results.

**Table 3. The statistical standard for determining the assessment levels of the quality of training programs for teachers of students with disabilities in light of recent trends in the current study.**

| Means | Degree |
|---|---|
| 1.00- < 1.80 | Very low |
| 1.80- < 2.60 | Low |
| 2.60- < 3.40 | Medium |
| 3.40- < 4.20 | High |
| 4.20–5.00 | Very high |

**Table 4. The means and standard deviations of the participants' responses on the assessment level of the quality of training programs for students with disabilities in light of modern trends.**

| No. | Domain | Means | Standard deviation | Rank | Level |
|---|---|---|---|---|---|
| 2 | Content selection and organization | 3.38 | 0.69 | 1 | Medium |
| 1 | Planning for education | 3.35 | 0.62 | 2 | Medium |
| 5 | Standards for acquiring scientific and technical knowledge | 3.35 | 0.68 | 3 | Medium |
| 3 | Methods and strategies for achieving goals and assessment | 3.28 | 0.61 | 4 | Medium |
| 4 | The role of the teacher and his various responsibilities | 3.24 | 0.61 | 5 | Medium |
| 6 | Standards for the effective use of educational skills | 3.22 | 0.74 | 6 | Medium |
| 7 | Standards for developing values and ideas | 3.19 | 0.72 | 7 | Medium |
| | Total | 3.29 | 0.51 | | Medium |

Table 4 shows that the total degree for the responses of the study sample on the quality assessment of the training programs for students with disabilities in light of modern trends scored (3.29) with a standard deviation of (0.51), and a medium degree. Also, it was shown the means of the domains of the study instrument ranged between (3.19–3.38), they all were medium. At domains, the second domain (content selection and organization) ranked first with a means of (3.38), a standard deviation of (0.69), and a medium level. Then, the first domain (planning for education) followed with a means of (3.35), a standard deviation (0.62), and a medium level. In the last place came the seventh domain (standards for developing values and ideas) with a means of (3.19), a standard deviation of (0.72), and a medium level.

Results of the second research question: *Are there statistically significant differences in the assessment level of the quality of training programs for teachers of students with disabilities in light of recent trends due to the gender variable*?

The t-test for two independent samples was used to show the differences between the means of the quality assessment of the training programs for students with disabilities in light of modern trends attributed to the gender variable. Table 5 depicts the results.

Table 5 shows no statistically significant differences at (0.05) between the means of the quality assessment of the training programs for students with disabilities in light of modern trends

**Table 5. T-test analysis for the quality assessment of the training programs for students with disabilities in light of modern trends attributed to gender.**

| Domain | Gender | No. | Means | Standard deviation | T | df | Sig. 2-talied |
|---|---|---|---|---|---|---|---|
| Planning for education | Male | 113 | 3.34 | 0.64 | .354 | 270 | .723 |
| | Female | 159 | 3.36 | 0.61 | | | |
| Content selection and organization | Male | 113 | 3.43 | 0.72 | 1.002 | 270 | .317 |
| | Female | 159 | 3.35 | 0.68 | | | |
| Methods and strategies for achieving goals and assessment | Male | 113 | 3.24 | 0.66 | .941 | 270 | .348 |
| | Female | 159 | 3.31 | 0.58 | | | |
| The role of the teacher and his various responsibilities | Male | 113 | 3.25 | 0.63 | .088 | 270 | .930 |
| | Female | 159 | 3.24 | 0.60 | | | |
| Standards for acquiring scientific and technical knowledge | Male | 113 | 3.35 | 0.70 | .058 | 270 | .954 |
| | Female | 159 | 3.35 | 0.66 | | | |
| Standards for the effective use of educational skills | Male | 113 | 3.19 | 0.80 | .582 | 270 | .561 |
| | Female | 159 | 3.24 | 0.69 | | | |
| Standards for developing values and ideas | Male | 113 | 3.19 | 0.80 | .046 | 270 | .964 |
| | Female | 159 | 3.20 | 0.66 | | | |
| Total | Male | 113 | 3.28 | 0.55 | .236 | 270 | .813 |
| | Female | 159 | 3.30 | 0.48 | | | |

**Table 6. T-test analysis for the quality assessment of the training programs for students with disabilities in light of modern trends attributed to academic qualification.**

| Domain | Qualification | No. | Means | Standard deviation | T | df | Sig. 2-talied |
|---|---|---|---|---|---|---|---|
| Planning for education | Bachelor | 182 | 3.27 | 0.64 | 3.117 | 270 | .002 |
| | Postgraduate studies | 90 | 3.52 | 0.56 | | | |
| Content selection and organization | Bachelor | 182 | 3.30 | 0.73 | 2.850 | 270 | .005 |
| | Postgraduate studies | 90 | 3.55 | 0.58 | | | |
| Methods and strategies for achieving goals and assessment | Bachelor | 182 | 3.21 | 0.65 | 2.726 | 270 | .007 |
| | Postgraduate studies | 90 | 3.43 | 0.52 | | | |
| The role of the teacher and his various responsibilities | Bachelor | 182 | 3.18 | 0.64 | 2.328 | 270 | .021 |
| | Postgraduate studies | 90 | 3.37 | 0.54 | | | |
| Standards for acquiring scientific and technical knowledge | Bachelor | 182 | 3.28 | 0.72 | 2.431 | 270 | .016 |
| | Postgraduate studies | 90 | 3.49 | 0.56 | | | |
| Standards for the effective use of educational skills | Bachelor | 182 | 3.13 | 0.75 | 2.951 | 270 | .003 |
| | Postgraduate studies | 90 | 3.40 | 0.67 | | | |
| Standards for developing values and ideas | Bachelor | 182 | 3.13 | 0.75 | 2.053 | 270 | .041 |
| | Postgraduate studies | 90 | 3.32 | 0.63 | | | |
| Total | Bachelor | 182 | 3.22 | 0.54 | 3.463 | 270 | .001 |
| | Postgraduate studies | 90 | 3.44 | 0.41 | | | |

due to the variable of gender on all domains and the whole scale. The values of the statistical significance were higher than (0.05).

Results of the third research question: *Are there statistically significant differences in the assessment level of the quality of training programs for teachers of students with disabilities in light of recent trends due to the variable of academic qualification*?

To answer the research question, the t-test for two independent samples was used to show the differences between the means of the quality assessment of the training programs for students with disabilities in light of modern trends attributed to the variable of academic qualification. Table 6 shows the results.

According to Table 6, there were statistically significant differences at (0.05) between the means of the quality assessment of the training programs for students with disabilities in light of modern trends attributed to academic qualification on all domains and the whole scale in favor of postgraduate studies. The values of the statistical significance were less than (0.05).

Results of the fourth research question: *Are there statistically significant differences in the assessment level of the quality of training programs for teachers of students with disabilities in light of recent trends due to the variable of years of experience*?

The researcher used the t-test for two independent samples to show the differences between the means of the quality assessment of the training programs for students with disabilities in light of modern trends attributed to the variable of years of experience. Table 7 displays the results.

Table 7 shows statistically significant differences at (0.05) between the means of the quality assessment of the training programs for students with disabilities in light of modern trends according to years of experience in all domains and the whole scale in favor of those with experience of more than 10 years. The values of the statistical significance were less than (0.05).

## Discussion

The first research question: The result of this question showed that the total degree of the study sample's responses on the quality assessment of the training programs for students with

**Table 7. T-test analysis for the quality assessment of the training programs for students with disabilities in light of modern trends attributed to years of experience.**

| Domain | Years of experience | No. | Means | Standard deviation | T | df | Sig. 2-talied |
|---|---|---|---|---|---|---|---|
| Planning for education | 1–10 years | 135 | 3.21 | 0.66 | 3.699 | 270 | .000 |
| | Above 10 years | 137 | 3.49 | 0.56 | | | |
| Content selection and organization | 1–10 years | 135 | 3.18 | 0.73 | 5.025 | 270 | .000 |
| | Above 10 years | 137 | 3.59 | 0.59 | | | |
| Methods and strategies for achieving goals and assessment | 1–10 years | 135 | 3.13 | 0.65 | 4.238 | 270 | .000 |
| | Above 10 years | 137 | 3.44 | 0.54 | | | |
| The role of the teacher and his various responsibilities | 1–10 years | 135 | 3.08 | 0.65 | 4.421 | 270 | .000 |
| | Above 10 years | 137 | 3.40 | 0.53 | | | |
| Standards for acquiring scientific and technical knowledge | 1–10 years | 135 | 3.14 | 0.72 | 5.330 | 270 | .000 |
| | Above 10 years | 137 | 3.56 | 0.56 | | | |
| Standards for the effective use of educational skills | 1–10 years | 135 | 3.11 | 0.71 | 2.400 | 270 | .017 |
| | Above 10 years | 137 | 3.32 | 0.74 | | | |
| Standards for developing values and ideas | 1–10 years | 135 | 3.08 | 0.76 | 2.592 | 270 | .010 |
| | Above 10 years | 137 | 3.31 | 0.66 | | | |
| Total | 1–10 years | 135 | 3.14 | 0.57 | 5.199 | 270 | .000 |
| | Above 10 years | 137 | 3.44 | 0.39 | | | |

disabilities in light of modern trends and domain came moderate. This refers to the disparity in the design of training programs for students with disabilities in light of modern trends and their focus on only some aspects. This may be due to the shortage of knowledge and information among the developers of these programs on modern trends in training teachers based on competency and technical and social approaches. Therefore, the training programs appeared somewhat incongruent with those approaches. Also, the result may be explained in light of the design of these programs that may not be highly based on the trainees' needs and opinions on the training requirements of the programs that keep pace with contemporary educational developments in the field of special education and care for students with disabilities. Hence, there must be a continuous assessment of these programs against the rapid developments in the field of special education in light of technological and knowledge expansion. Therefore, educational institutions are required to develop themselves and their programs in line with these rapid changes in quantity and quality in the way their programs are presented.

The second research question: The result of this question showed that there were no statistically significant differences at (0.05) between the means of the assessment level of the quality of training programs for teachers of students with disabilities in light of recent trends according to the gender variable in all domains and the total score. This may be due to the teachers (males and females) having received the same training programs for special education teachers (teachers of students with disabilities) launched by the Ministry of Education in the Kingdom of Saudi Arabia. They had the same training conditions and environments as well as the same training contents they were exposed to. These factors resulted in the absence of statistical differences in assessing the quality of those programs.

The third research question: The results of this question showed statistically significant differences at (0.05) between the means of the assessment level of the quality of training programs for the teachers of students with disabilities in light of recent trends in all the domains and the total score of the administered scale according to the academic qualification in favor of postgraduate studies. This can be attributed to the fact that teachers with postgraduate qualifications may have possessed information, knowledge, and facts related to the quality of training programs for the teachers of students with disabilities in light of recent trends as a result of the

intensive study in the postgraduate program about the contemporary competencies they need to work with students with disabilities. This has affected the extent of their awareness and clarity of knowledge and their accurate assessment of the quality of training programs for the teachers of students with disabilities in light of recent trends. Accordingly, it is important that the teachers of students with disabilities, who have academic qualifications in postgraduate studies, review the training programs when they are planned and managed. They share their opinion about keeping pace with these programs for their needs in light of modern trends and models. So, they are fully prepared and take into their account all aspects of those training programs.

The fourth research question: The result of this question showed statistically significant differences at (0.05) between the means of the assessment level of the quality of training programs for the teachers of students with disabilities in light of recent trends in all the domains and the total score according to the variable of years of experience in favor of the category (above 10 years). This may be because the teachers with more experience have acquired information and facts related to the quality of training programs for teachers of students with disabilities in light of recent trends due to of their attendance and participation in several training programs and courses during their years of work. This allowed them to form trends and a clear picture of the quality of these programs and their connection to the competency approach, technical and social approach. This may also be attributed to teachers with longer experience having a more in-depth, diverse, and cultural view of the quality of training programs and what meets their needs and the needs of their students with disabilities. This is reflected in their knowledge and information on how to assess the training programs offered to them in the light of recent trends.

## Conclusion

The current study shed light on the level of assessment of the quality of training programs for teachers of students with disabilities concerning recent trends in the Kingdom of Saudi Arabia. This study was determined in light of its quantitative and descriptive topic of the phenomenon of quality of training programs for teachers of students with disabilities in recent trends from the teachers' point of view and the differences in their point according to gender, academic qualification and years of experience. An attempt has been made as much as possible to analyze and detail the level of assessment of the quality of training programs for teachers of students with disabilities in light of recent trends in the Kingdom of Saudi Arabia. There was a good level of implementation of these programs in light of recent trends on the ground in the fields of content selection and organization and helping the teacher to plan education, and standards for acquiring scientific and technical knowledge. There were other areas in which there was a need for improvement and development such as the field of the teacher's role and various responsibilities, standards for the effective use of educational skills and the development of values and ideas. Here perfection is sought in that to keep pace with global institutions, achieve the required competition and excel in later stages. The results of the study revealed that the overall degree to assess the quality of training programs for teachers of students with disabilities in light of recent trends from the point of view of the study sample was medium. Also, there were statistically significant differences in the assessment degree of the quality of training programs for teachers of students with disabilities in light of recent trends due to the variables of academic qualification in favor of postgraduate studies, and years of experience in favor of those who had more than 10 years of experience. However, no statistically significant differences were shown due to the gender variable.

This study implies the important role of training programs in improving teachers' knowledge, experiences and attitudes. Also, these programs help teachers update and develop

their information, raise their efficiency, and improve their teaching and job performance, which is reflected on students with disabilities. In addition, their levels and abilities are boosted in line with educational and modern technological developments to achieve a competitive advantage in education.

In light of the results of the study, the study recommended that the Ministry of Education, its departments and institutions, and those responsible for setting educational policies should pay attention to the adoption of a strategic plan to improve the quality of training programs for teachers of students with disabilities by relying on modern models and trends in teacher training. Also, the General Administration of Special Education in the Kingdom of Saudi Arabia needs to adopt a strategic plan that meets the needs of trainees of teachers of students with disabilities. The content of training programs should be planned and selected to meet the needs of trainees in light of recent trends in training in general and in the field of standards for acquiring scientific and technical knowledge, standards for the effective use of educational skills, and standards for developing values and ideas in particular. In addition, officials and supervisors in training programs for teachers of students with disabilities should select highly qualified trainers who can design training programs in light of recent trends. Future studies similar to this study may focus on other models and theories of recent trends in teacher training or relying on training standards for teachers issued by international development and quality bodies and organizations in developed countries such as the United States of America, Britain and Japan. Also, the opinions of school principals, educational supervisors and training and delegation officials in the education directorates may be surveyed. In addition, it is important to conduct exploratory studies that show the impact of training programs offered to teachers of students with disabilities on their teaching and professional performance. Finally, there is a need to conduct a future diagnostic study of the most important challenges and obstacles and proposed solutions to the quality of training programs for teachers of students with disabilities.

## Author Contributions

**Conceptualization:** Rasheed Khuwayshan Algethami.

**Data curation:** Rasheed Khuwayshan Algethami.

**Formal analysis:** Rasheed Khuwayshan Algethami.

**Funding acquisition:** Rasheed Khuwayshan Algethami.

**Investigation:** Rasheed Khuwayshan Algethami.

**Methodology:** Rasheed Khuwayshan Algethami.

**Project administration:** Rasheed Khuwayshan Algethami.

**Resources:** Rasheed Khuwayshan Algethami.

**Software:** Rasheed Khuwayshan Algethami.

**Supervision:** Rasheed Khuwayshan Algethami.

**Validation:** Rasheed Khuwayshan Algethami.

**Visualization:** Rasheed Khuwayshan Algethami.

**Writing – original draft:** Rasheed Khuwayshan Algethami.

**Writing – review & editing:** Rasheed Khuwayshan Algethami.

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
