## [Decision Letter · Decision Letter 0]

5 May 2022

PONE-D-22-12040Assessing the Quality of Training Programs for Teachers of Students with Disabilities in Light of Recent Trends in the Kingdom of Saudi ArabiaPLOS ONE

Dear Dr. ALZUBI,

Thank you for submitting your manuscript to PLOS ONE. After careful consideration, we feel that it has merit but does not fully meet PLOS ONE’s publication criteria as it currently stands. Therefore, we invite you to submit a revised version of the manuscript that addresses the points raised during the review process.

We look forward to receiving your revised manuscript.

Kind regards,

Sathishkumar V E

Academic Editor

PLOS ONE

Journal Requirements:

3. Please ensure that you include a title page within your main document. We do appreciate that you have a title page document uploaded as a separate file, however, as per our author guidelines (http://journals.plos.org/plosone/s/submission-guidelines#loc-title-page) we do require this to be part of the manuscript file itself and not uploaded separately.

"The author is thankful to the Deanship of Scientific Research at Taif

University for the financial and technical support of this research (17-1422-1)"

Reviewers' comments:

Reviewer's Responses to Questions

**Comments to the Author**

1. Is the manuscript technically sound, and do the data support the conclusions?

Reviewer #1: Yes

Reviewer #2: Yes

Reviewer #3: Yes

2. Has the statistical analysis been performed appropriately and rigorously? 

Reviewer #1: Yes

Reviewer #2: Yes

Reviewer #3: Yes

3. Have the authors made all data underlying the findings in their manuscript fully available?

Reviewer #1: Yes

Reviewer #2: Yes

Reviewer #3: Yes

4. Is the manuscript presented in an intelligible fashion and written in standard English?

Reviewer #1: Yes

Reviewer #2: No

Reviewer #3: Yes

5. Review Comments to the Author

Reviewer #1: Very interesting topic. Good job

Reviewer #2: The Research Paper needs the following revisions and is subject for re-review, and after re-review, the final decision for the manuscript will be done:

1. Add in the last lines of abstract, what observations are there in the study.

2. Add more towards scope of the problem in Introduction section.

3. Add Objectives of the paper at end of Introduction. Add organization of the paper.

4. Add more information towards model formulation and how the statistical analysis is conducted.

5. Add future scope to the study.

Reviewer #3: Introduction and Background study can be made as a separate section

Overall presentation is good

In Discussion section why Second research question topic is repeated?

A complete language check should be done

Conclusion part should be added to summarze the manuscript results

6. PLOS authors have the option to publish the peer review history of their article (what does this mean?). If published, this will include your full peer review and any attached files.

Reviewer #1: **Yes: **Majed Alamri

Reviewer #2: **Yes: **Anand Nayyar

Reviewer #3: **Yes: **Usha Moorthy

---

## [Author Response · Author response to Decision Letter 0]

27 May 2022

Response to Reviewers

Reviewer #2

1. Add in the last lines of Abstract. What observations are in the study? 

There were weaknesses in the training programs in terms of training teachers on their various roles and responsibilities, standards for the effective use of educational skills, and the development of values and ideas. Also, these training programs did not take into account the development of the values associated with technology and its relationship to teachers. In light of the results, the study recommended that the Ministry of Education, its departments and institutions, and those responsible for setting educational policies should pay attention to the adoption of a strategic plan to improve the quality of training programs for teachers of students with disabilities by relying on modern models and trends in teacher training. There is a need to conduct a future diagnostic study of the most important challenges and obstacles and proposed solutions to the quality of training programs for teachers of students with disabilities.

2. Add more towards scope of the problem in introduction section.

Training and professional development for teachers of students with disabilities are among the modern approaches that occupy educational institutions and those interested in reform issues in special education at present. They contribute to enhancing the knowledge, skills, and experiences of teachers in various fields emotionally and professionally so that they do not stop at the level they reached upon graduation or employment in the education sector (Lozano & Yildiz, 2015).

In recent years, the term (Quality) has spread. It is a modern management philosophy based on a number of modern directed management concepts on which basic administrative means and creative efforts are mixed with specialized technical skills to raise the level of performance and continuously improve and develop institutions and human resources. Therefore, there have been voices calling for quality in the education sector similar to other sectors. It was a turning point in the path of educational reform through the presence of a team with extensive experience to lead this movement and change and achieve the objectives of improving educational quality in pre-service and in-service teacher training programs (Al-Rushdi, 2013).

Quality is of great importance in determining the basic pillars in the framework of practical application in public education institutions. These pillars would indicate the basic facts that should be relied upon in the field of quality assurance and quality control to apply advanced methods to ensure their standards as well as continuous improvement and achieve the highest possible levels in educational practices and processes. Thus, the outputs of educational institutions are improved. One of the most important of these pillars is training and professional development for teachers (Ghosh, 2020).

Therefore, the researcher believes that the issue of quality in training programs for teachers constitutes a challenge for public and private education institutions. The initiative to raise this topic is important in drawing the attention of those in charge of training programs seriously. It is necessary to continuously strive to develop teachers’ skills, including teachers of students with disabilities scientifically and professionally. The continuous development of teachers’ skills will keep pace with the era of innovation, spread the culture of training as a basis for the culture of entrepreneurship, and develop training strategies in line with modern global trends in training.

3. Add objectives of the paper at the end of introduction. Add organization of the paper. 

Objectives of the study

This study aimed to determine the level of assessment of the quality of training programs for teachers of students with disabilities in light of recent trends to work for their development and enhancement to effectively achieve their desired goals. Also, the study aimed to unveil the statistically significant differences in the study sample’s responses about whether the level of assessment of the quality of training programs for teachers of students with disabilities in light of recent trends is due to the variables of gender, academic qualification, and years of experience to see their views and develop the quality level of training programs for teachers of students with disabilities in line with recent trends to achieve competitive advantage.

4. Add more information towards model formulation and how the statistical analysis is conducted. 

This study used the statistical software of (SPSS) version (23) to analyze the data of the study and answer the research questions. Pearson correlation coefficient to verify the consistency of the instrument and Cronbach’s alpha coefficient to verify the reliability of the study instrument. Also, means, standard deviations, and ranks were calculated to answer the first question to identify the level of assessment of the quality of training programs for teachers of students with disabilities in light of recent trends from the point of view of the study sample. In addition, the t-test was used to answer the second, third, and fourth questions to show the significance of the differences between the means for the level of quality assessment of training programs for teachers of students with disabilities in light of recent trends from the point of view of the study sample according to demographic variables (gender, academic qualification, and years of experience).

5. Add future scope to the study

Future studies similar to this study may focus on other models and theories of recent trends in teacher training or relying on training standards for teachers issued by international development and quality bodies and organizations in developed countries such as the United States of America, Britain and Japan. Also, the opinions of school principals, educational supervisors and training and delegation officials in the education directorates may be surveyed. In addition, it is important to conduct exploratory studies that show the impact of training programs offered to teachers of students with disabilities on their teaching and professional performance. Finally, there is a need to conduct a future diagnostic study of the most important challenges and obstacles and proposed solutions to the quality of training programs for teachers of students with disabilities.

Reviewer #3

1. Introduction and background study can be made as a spate section. 

Introduction: 

Training and professional development for teachers of students with disabilities are among the modern approaches that occupy educational institutions and those interested in reform issues in special education at present. They contribute to enhancing the knowledge, skills, and experiences of teachers in various fields emotionally and professionally so that they do not stop at the level they reached upon graduation or employment in the education sector (Lozano & Yildiz, 2015).

In recent years, the term (Quality) has spread. It is a modern management philosophy based on a number of modern directed management concepts on which basic administrative means and creative efforts are mixed with specialized technical skills to raise the level of performance and continuously improve and develop institutions and human resources. Therefore, there have been voices calling for quality in the education sector similar to other sectors. It was a turning point in the path of educational reform through the presence of a team with extensive experience to lead this movement and change and achieve the objectives of improving educational quality in pre-service and in-service teacher training programs (Al-Rushdi, 2013).

Quality is of great importance in determining the basic pillars in the framework of practical application in public education institutions. These pillars would indicate the basic facts that should be relied upon in the field of quality assurance and quality control to apply advanced methods to ensure their standards as well as continuous improvement and achieve the highest possible levels in educational practices and processes. Thus, the outputs of educational institutions are improved. One of the most important of these pillars is training and professional development for teachers (Ghosh, 2020).

Therefore, the researcher believes that the issue of quality in training programs for teachers constitutes a challenge for public and private education institutions. The initiative to raise this topic is important in drawing the attention of those in charge of training programs seriously. It is necessary to continuously strive to develop teachers’ skills, including teachers of students with disabilities scientifically and professionally. The continuous development of teachers’ skills will keep pace with the era of innovation, spread the culture of training as a basis for the culture of entrepreneurship, and develop training strategies in line with modern global trends in training.

Background:

Human resources have received great attention from departments and institutions because of the impact they have on the success of these bodies in various administrative, social, and economic fields. Therefore, the issue of developing the competencies of human resources in these bodies and institutions is one of the matters that has become imperative for their advancement and the services they provide to society. Also, the role of training is considered one of the most important processes in the development of human resources in these bodies and institutions (Balbay, Pamuk, Temir, & Dogan, 2018). The training philosophy is based on a clear logic that assumes that the required numbers and types of human competencies to practice certain jobs require a high level of competence, as is the case for the required disciplines (Brown & Atkins, 2016). The philosophy of training is based on a basic rule that even in the case of academic institutions from which specialists in various fields of work graduate, they will still need training. Training programs provide methods and open new horizons of professional and scientific knowledge for the practitioner, whose theoretical study allows him to interact with them. In addition, the practitioner is in constant need to renew his information, develop existing knowledge, and stand up to the latest methods and theories in the field of specialization. Organized training programs provide the above-mentioned skills to PR actioners (Gegenfurtner, 2019). On the approach of the various community institutions, the educational institutions proceeded and devoted attention to training their employees, especially teachers, including teachers of students with disabilities (So et al., 2021). This came as a result of the importance of professional development and professional growth to provide the teacher with various knowledge, skills, and experiences to keep pace with modern educational developments in their field of specialization in light of their needs (Byrd & Alexander, 2020). In-service teacher training is an important factor for developing their performance and increasing productivity. Therefore, it is considered an investment expenditure that achieves a tangible return to contribute to meeting the needs of the economic and social growth of the nation. Training represents the satisfaction of their psychological and cognitive needs, which is reflected in the mastery of their work for the sake of professional growth (Zine El Abidine, 2021).

Teacher preparation and training programs are important in their professional development and growth. They are planned and organized programs according to educational and psychological theories. Also, they are carried out by specialized educational institutions to provide teachers with scientific and applied experiences that enable them to grow in the teaching profession and increase their educational production (Ghosh, 2020).

Interest in developing teacher training programs and ensuring their quality is one of the most important contemporary global trends. They receive increasing attention with the different degrees of focus in development processes depending on the societal context from one country to another. Interest in teacher preparation and development has become a political priority in most countries of the world (Lodhi & Ghias, 2019). In developed countries, there is an increasing interest in ensuring the provision of sufficient and required numbers of teachers to meet the increasing retirement cases while ensuring the quality of preparation. In developing countries, we find an increasing need to raise the level of qualifications of teachers with attention to quality. In light of contemporary educational trends and the emergence of new patterns and methods in teaching, it becomes clear the urgent need for a change in the future roles of the teacher. Consequently, there is a review of teacher preparation and training programs in light of contemporary roles and challenges. For these programs to be effective, they require development in their objectives, mechanisms, and methods to overcome the current weaknesses (Masadeh, 2012).

Hence, educational institutions seek to reach the quality of service and the scientific and professional levels of the teacher by improving his educational, research, and career performance by setting up special training programs. This can be implemented and supervised by experienced and specialized competencies in the field of training. As a result, the interest in training programs for teachers has emerged. Training programs and courses give teachers new ways that facilitate the teaching process and help them to overcome the obstacles created by the development of the educational field. Also, they facilitate the effort made to communicate ideas to students and ways to memorize and review them (Jackson, 2021). Therefore, training and professional development programs for teachers must include various areas that include mastering modern teaching strategies and enhancing teachers’ competencies in preparing lessons in a manner commensurate with the needs of learners. Also, they include using modern technology in the teaching process in the classroom (Ghosh, 2020). The importance of teacher training programs including teachers of students with disabilities and recent trends has emerged and aimed at achieving teachers’ competencies and raising their level of performance in this changing world. They also emphasize the continuity of this performance effectively when working professionally. In addition, some of these trends emphasize practical aspects and give teachers special weight in preparation and training programs (Cornelius, Rosenberg, & Sandmel, 2020). Hence the need for educational systems to consider developing programs to train teachers of people with disabilities continuously to keep pace with the rapid civilizational developments taking place in the era (Byrd & Alexander, 2020). Therefore, several trends have emerged in teacher training programs focusing on different models, approaches, processes, and competencies (Crispel & Kasperski, 2019). Some of these approaches are based on competency. It focuses on the importance of training teachers to possess skills such as planning education, selecting and organizing content, using methods and means that achieve goals and assessment, the teacher’s role, and various responsibilities as a guide and leader. Another approach includes teacher-training programs that are based on science, technology, and society. Bybee (cited in Ben Zayan, 2018) provided a conceptual framework for the knowledge, skills, and values of scientific and technical enlightenment. He defines a set of standards that can be used in building training programs such as standards for acquiring scientific and technical knowledge by studying life matters, and contemporary social issues. The standards also include standards for the effective use of educational skills that depend on scientific and technical investigation, participation in information gathering and problem-solving, and standards for developing values and ideas related to science and technology and their relationship to society. The current study focused on these two approaches because they are among the most recent approaches that keep pace with the present era and its educational, scientific, and technical developments in the field of special education and teacher preparation and training.

The teacher of students with disabilities is an important hub in the teaching and learning processes of individuals with disabilities. He is responsible for implementing the individual educational plan, improving and developing their knowledge and skills, and following up on their achievements and performance (Athanasiadis & Syriopoulou-Delli, 2010). Likewise, he is the main executor of the individual educational program, follow-up to the family plan, the guide for students with disabilities in the processes of teaching and learning, and responsible for following up on their academic achievement and solving their problems (Young, 2018). Training teachers of students with disabilities is one of the issues that concern educational officials who believe in their great role in educating students with disabilities. Also, they refine their skills to perform well in teaching students with disabilities and to keep pace with the scientific and technical development in the education of students with disabilities (Hester, Bridges, & Rollins, 2020). Hence, specifically in the summer of 2019 and the summer of 2020, the Kingdom of Saudi Arabia, represented by the Ministry of Education and the General Administration of Special Education, in cooperation with a group of companies, associations, and experts in the field of training and human development, and in various public educational areas, launched an initiative to train special education teachers (teachers of students with disabilities). Hence, specifically in the summers of 2019 and 2020, the Kingdom of Saudi Arabia, represented by the Ministry of Education and the General Department of Special Education in cooperation with a group of companies, associations, and experts in the field of training and human development, and in various public educational areas, launched an initiative to train special education teachers (teachers of students with disabilities). The initiative aimed to provide teachers with strategies, teaching methods appropriate for teaching students with disabilities according to categories and grades, discovering problems among children with disabilities, and ways to deal with them. Also, it aimed to train teachers on using appropriate assistive technologies in teaching and learning students with disabilities, monitoring practices based on evidence and scientific evidence in the field of disability, and teaching methods of students with disabilities in light of the COVID-19 pandemic, and how to create an appropriate psychological atmosphere for children with disabilities using storytelling and social story (Saudi Ministry of Education, 2019-2020).

As mentioned above, the significance of this study is evident by focusing on an important topic of management and human resources in educational institutions, which is the issue of the quality of training programs for teachers of students with disabilities in light of recent trends.

2. In discussion section why second research question topic is repeated? 

The second research question in the discussion section was repeated by mistake. It has been fixed. No repetition is there now. You may check the main manuscript. 

3. A complete language check should be done

The language of the whole manuscript has been checked and proofread. 

4. Conclusion part should be added to summarize the manuscript results 

The current study shed light on the level of assessment of the quality of training programs for teachers of students with disabilities concerning recent trends in the Kingdom of Saudi Arabia. This study was determined in light of its quantitative and descriptive topic of the phenomenon of quality of training programs for teachers of students with disabilities in recent trends from the teachers’ point of view and the differences in their point according to gender, academic qualification and years of experience. An attempt has been made as much as possible to analyze and detail the level of assessment of the quality of training programs for teachers of students with disabilities in light of recent trends in the Kingdom of Saudi Arabia. There was a good level of implementation of these programs in light of recent trends on the ground in the fields of content selection and organization and helping the teacher to plan education, and standards for acquiring scientific and technical knowledge. There were other areas in which there was a need for improvement and development such as the field of the teacher’s role and various responsibilities, standards for the effective use of educational skills and the development of values and ideas. Here perfection is sought in that to keep pace with global institutions, achieve the required competition and excel in later stages. The results of the study revealed that the overall degree to assess the quality of training programs for teachers of students with disabilities in light of recent trends from the point of view of the study sample was medium. Also, there were statistically significant differences in the assessment degree of the quality of training programs for teachers of students with disabilities in light of recent trends due to the variables of academic qualification in favor of postgraduate studies, and years of experience in favor of those who had more than 10 years of experience. However, no statistically significant differences were shown due to the gender variable.

This study implicates the important role of training programs in improving teachers’ knowledge, experiences and attitudes. Also, they help teachers update and develop their information, raise their efficiency, and improve their teaching and job performance, which is reflected on students with disabilities. In addition, their levels and abilities are boosted in line with educational and modern technological developments to achieve a competitive advantage in education. In light of the results of the study, the study recommended that the Ministry of Education, its departments and institutions, and those responsible for setting educational policies should pay attention to the adoption of a strategic plan to improve the quality of training programs for teachers of students with disabilities by relying on modern models and trends in teacher training. Also, the General Administration of Special Education in the Kingdom of Saudi Arabia needs to adopt a strategic plan that meets the needs of trainees of teachers of students with disabilities. The content of training programs should be planned and selected to meet the needs of trainees in light of recent trends in training in general and in the field of standards for acquiring scientific and technical knowledge, standards for the effective use of educational skills, and standards for developing values and ideas in particular. In addition, officials and supervisors in training programs for teachers of students with disabilities should select highly qualified trainers who can design training programs in light of recent trends. Future studies similar to this study may focus on other models and theories of recent trends in teacher training or relying on training standards for teachers issued by international development and quality bodies and organizations in developed countries such as the United States of America, Britain and Japan. Also, the opinions of school principals, educational supervisors and training and delegation officials in the education directorates may be surveyed. In addition, it is important to conduct exploratory studies that show the impact of training programs offered to teachers of students with disabilities on their teaching and professional performance. Finally, there is a need to conduct a future diagnostic study of the most important challenges and obstacles and proposed solutions to the quality of training programs for teachers of students with disabilities.

---

## [Decision Letter · Decision Letter 1]

12 Jun 2022

Assessing the Quality of Training Programs for Teachers of Students with Disabilities in Light of Recent Trends in the Kingdom of Saudi Arabia

PONE-D-22-12040R1

Dear Dr. ALZUBI,

We’re pleased to inform you that your manuscript has been judged scientifically suitable for publication and will be formally accepted for publication once it meets all outstanding technical requirements.

Kind regards,

Sathishkumar V E

Academic Editor

PLOS ONE

Additional Editor Comments (optional):

Reviewers' comments:

Reviewer's Responses to Questions

**Comments to the Author**

1. If the authors have adequately addressed your comments raised in a previous round of review and you feel that this manuscript is now acceptable for publication, you may indicate that here to bypass the “Comments to the Author” section, enter your conflict of interest statement in the “Confidential to Editor” section, and submit your "Accept" recommendation.

Reviewer #1: All comments have been addressed

Reviewer #3: (No Response)

2. Is the manuscript technically sound, and do the data support the conclusions?

Reviewer #1: Yes

Reviewer #3: (No Response)

3. Has the statistical analysis been performed appropriately and rigorously? 

Reviewer #1: Yes

Reviewer #3: (No Response)

4. Have the authors made all data underlying the findings in their manuscript fully available?

Reviewer #1: Yes

Reviewer #3: (No Response)

5. Is the manuscript presented in an intelligible fashion and written in standard English?

Reviewer #1: Yes

Reviewer #3: (No Response)

6. Review Comments to the Author

Reviewer #1: (No Response)

Reviewer #3: (No Response)

7. PLOS authors have the option to publish the peer review history of their article (what does this mean?). If published, this will include your full peer review and any attached files.

Reviewer #1: No

Reviewer #3: **Yes: **Usha Moorthy

---

## [Editor Report · Acceptance letter]

30 Jun 2022

PONE-D-22-12040R1 

Assessing the Quality of Training Programs for Teachers of Students with Disabilities in Light of Recent Trends in the Kingdom of Saudi Arabia 

Dear Dr. Algethami:

I'm pleased to inform you that your manuscript has been deemed suitable for publication in PLOS ONE. Congratulations! Your manuscript is now with our production department. 

Kind regards, 

on behalf of

Dr. Sathishkumar V E 

Academic Editor

PLOS ONE